A genetic programming-based ensemble method for long-term electricity demand forecasting

Issa Hayat Ahmed 1
Çevik Hasan Hüseyin 2
Yilmaz Ahmet 3
http://orcid.org/0000-0002-5031-7618 Cunkas Mehmet 2 mcunkas@selcuk.edu.tr
1 Electrical & Electronics Engineering, Institute of Science, Selcuk University , Selcuklu, Konya , Türkiye
2 Electrical & Electronics Engineering, Faculty of Technology, Selcuk University , Selcuklu, Konya , Türkiye
3 Computer Engineering, Faculty of Engineering, Karamanoglu Mehmetbey University , Karaman , Türkiye
Pamucar Dragan
Electronic publication date: 2025 Apr 16
Publication date: 2025
Volume: 11
Electronic Location ID: e2825
Received 2024 Oct 22; Accepted 2025 Mar 23
Copyright: © 2025 Issa et al.
Copyright year: 2025
Copyright holder: Issa et al.
License: This is an open access article distributed under the terms of the Creative Commons Attribution License, which permits unrestricted use, distribution, reproduction and adaptation in any medium and for any purpose provided that it is properly attributed. For attribution, the original author(s), title, publication source (PeerJ Computer Science) and either DOI or URL of the article must be cited.
License URL: https://creativecommons.org/licenses/by/4.0/

Keywords: Ensemble method, Long-term electricity demand, Genetic programming, Genetic algorithms, Particle swarm optimization, Simulated annealing

Funding: The authors received no funding for this work.

==============================
This study introduces a novel genetic programming-based ensemble method for forecasting long-term electricity consumption in Ethiopia. The technique utilizes a two-stage ensemble approach to project Ethiopia’s electricity consumption through 2031. In the initial stage, genetic algorithms, particle swarm optimization, and simulated annealing methods are applied to various regression models (linear, quadratic, and exponential). The preliminary forecast values generated in this stage were further refined in the second stage. Here, the genetic programming method was utilized to develop a formula based on the initial forecast values, which then provided the final forecast results. The most accurate predictions in the first stage were obtained using the GA_Quadratic, PSO_Quadratic, and SA_Quadratic methods, resulting in mean absolute percentage error (MAPE) values of 3.61, 3.63, and 4.68, respectively. In the second stage, the GP-based prediction achieved an even lower MAPE value of 2.83. Other error metrics, including MSE, root mean square error (RMSE), and R2, were also evaluated, with the proposed model outperforming all methods from the first stage on these metrics. The study projected Ethiopia’s total annual electricity consumption through 2031 under two different scenarios. Both scenarios indicate that by 2031, electricity consumption will have tripled compared to 2021 levels.

Introduction

The consumption of electricity is closely tied to a country’s level of economic development, with substantial evidence showing a causal relationship between economic growth and electricity consumption. In developed nations, electricity use is high and consistently increasing, while in developing countries, rapid economic expansion results in volatile macroeconomic variables, causing significant fluctuations in electricity demand. Consequently, accurately forecasting electricity consumption is crucial for effective energy planning, strategy development, and formulating energy policies that consider future economic growth, which is one of the most critical and complex challenges faced by electric utilities (Belke, Dobnik & Dreger, 2011; Kaboli, Selvaraj & Rahim, 2016).

Ethiopia, the second most populous nation in Africa after Nigeria, has a population exceeding 120 million. A large majority of Ethiopians live in rural areas and are primarily engaged in agricultural activities. However, recent years have seen a marked trend toward urbanization as more individuals migrate to cities in search of better economic opportunities. The Ethiopian population is notably young, with a significant portion under the age of 30. This youthful demographic presents both opportunities and challenges: it provides a dynamic source of energy and potential for driving economic growth and social progress, yet simultaneously presents challenges related to unemployment and access to education and healthcare. Ethiopia’s relatively high population growth rate further complicates the government’s efforts to provide adequate services and infrastructure to meet the needs of its growing populace (Embassy of Ethiopia, 2020; Liyew, Ejigu & Habtu, 2024; Mohammed & Yıldırım, 2023; Wikipidia, 2024). The government has implemented policies to attract foreign investment and promote the development of industrial parks and manufacturing sectors. There has been significant growth in the textile, apparel, and leather goods industries. Infrastructure development is another primary focus for the Ethiopian government, with investments in roads, railways, airports, and energy systems. Despite these positive developments, Ethiopia faces challenges such as high levels of poverty, unemployment, and income inequality (Embassy of Ethiopia, 2020; Gebrehiwot, 2021; Oqubay, 2019; Wikipidia, 2024). In recent years, Ethiopia has seen a significant rise in electrical energy demand, increasing by 30% over the past 5 years, driven by its rapidly growing economy and an increasing number of consumers. Despite this rising demand, the current electrical energy supply is inadequate, leaving most households without access to electricity and with limited availability of backup generators. To tackle these challenges, robust and appropriate load forecasting methods must guide energy planning and supply (Embassy of Ethiopia, 2020; Issa, 2024).

There are various studies on long-term load forecasting in the literature. The methods used include fuzzy logic (Ali et al., 2016), particle swarm optimization (PSO) (AlRashidi & EL-Naggar, 2010), genetic algorithms (GA) (Santra & Lin, 2019), artificial neural network (ANN) (Çunkaş & Altun, 2010), simulating annealing (SA), artificial cooperative search algorithm (Kaboli, Selvaraj & Rahim, 2016), genetic programming (GP) (Çunkaş & Taşkiran, 2011; Karabulut, Alkan & Yilmaz, 2008), and multi-objective optimization (Yang et al., 2024b). Some studies use a single estimation method, while others employ multiple methods for comparative analysis. Additionally, several investigations have explored hybrid approaches that combine various techniques to enhance forecasting accuracy. The field of forecasting is inherently dynamic, necessitating continual refinement as the underlying data are perpetually updated, new parameters influencing predictions may emerge, and novel forecasting models are developed. Furthermore, no universally superior method exists for all contexts and datasets; thus, researchers are engaged in an ongoing quest to identify the most effective models tailored to the specific characteristics of the data.

When examining the extant literature on Ethiopia, Gebremeskel, Ahlgren & Beyene (2021) employed the Long-range Energy Alternative Planning (LEAP) system to project the nation’s long-term energy requirements. Their study covers a 33-year planning horizon, from 2018 to 2050, and predicts a significant increase in energy consumption, particularly in electricity demand, driven by economic and population growth. Similarly, Lemma, Fenta & Vadlamudi (2022) developed a neural network-based model for long-term peak load forecasting within the Southern Ethiopian power grid. The model’s input variables include the number of domestic and industrial customers, gross domestic product (GDP), and commercial customers. Degefa (2022) constructed a gray forecasting model to predict Ethiopia’s future energy consumption. The model, trained on actual energy consumption data from 2008 to 2017, was applied to various energy types, including electricity, petroleum products, solid biomass, and total energy consumption. Gebreyohans, Saxena & Kumar (2018) carried out an extensive study on long-term load forecasting for the city of Wolayta Sodo, Ethiopia, employing hybrid models that integrate multivariate linear regression with artificial neural networks, as well as multivariate linear regression with an adaptive neuro-fuzzy inference system. Matewose (2016) focused on energy forecasting for the city of Jimma, Ethiopia. This research utilized monthly and accurate yearly load data obtained from the Jimma distribution system substation as a case study, comparing linear, compound growth, and quadratic regression models.

When analyzing energy forecasting studies related to Ethiopia, it is noted that some focus on predictions for a single city or region, some are limited to peak electricity load forecasts, and others consider energy consumption for heating and transportation. Although there are studies estimating Ethiopia’s annual electricity consumption, they are few, and many are not based on updated forecasts. The need to continuously update annual energy forecasts underscores the necessity for new predictions in this area.

Ensemble learning methods have gained significant traction in machine learning over recent years due to their superior predictive performance (Fan et al., 2024; Li et al., 2024; Xiao, Fang & Wang, 2024). These methods combine the predictions of multiple models to produce a more accurate outcome. Various algorithms can serve as base learners, including linear and nonlinear models, support vector machines, ANN, and other traditional machine learning techniques. By aggregating these models, ensemble methods effectively reduce the variance inherent in individual algorithms, thereby enhancing predictive performance accuracy. The initial stage involves generating forecasts using various individual models. The predictions from these models are then assessed, and the good performers are chosen to enhance accuracy in the next stage. In the second stage, further processing is applied to these selected forecasts. This process may include the average, weighted average, or more complex derivatives of the values predicted by the models used in the first stage to refine the final estimates. Ensemble methods facilitate the integration of different forecasting techniques, such as machine learning models, statistical approaches, and optimization strategies, leveraging their combined strengths and performance.

Among ensemble techniques, decision tree-based models such as extreme gradient boosting (XGBoost) (Kumar, Sood & Rawat, 2023) and Random Forest (Hassan et al., 2023) have been extensively utilized. Recent research has investigated the integration of GP to create nonlinear nodes within decision trees, yielding promising improvements in both interpretability and accuracy (Zhang et al., 2023). An Ensemble method employing GP for seasonal precipitation prediction has been proposed (Danandeh Mehr, 2020). This model utilized predictions generated by GP, gene expression programming, and multiple regression methods as inputs for a subsequent GP phase. The results indicated that the proposed model could enhance the forecasting accuracy of the best standalone models by up to 30%. Additionally, ensemble GP has demonstrated superior performance compared to individual methods in addressing the Dynamic Flexible Job Shop Scheduling problem (Xu et al., 2023) and in various classification problems (Meng et al., 2024).

As stated above, the studies aiming at energy forecasting for Ethiopia offer valuable contributions. However, it is essential to note that, to the authors’ knowledge, no existing research has yet utilized various optimization methods to identify which approaches are most efficient for forecasting Ethiopia’s energy needs demand.

This study seeks to answer two fundamental questions: 1) Is it possible to accurately forecast Ethiopia’s energy demand employing optimization techniques such as GA, PSO, and SA using the population, GDP, and import and export data?

2) Can the results obtained with GA, PSO, and SA be further improved using a GP-based ensemble method?

In this study, Ethiopia’s electricity demand is estimated using a novel ensemble method based on GP. This model combines the features of various approaches for the nonlinear prediction of complex data, often providing more stability and accuracy than any single model. Electricity demand forecasting is conducted using four methods: GA, PSO, SA, and GP. Their performances are then compared to each other and conventional regression methods. In the first phase, GA, PSO, and SA optimization techniques are applied to linear, quadratic, and exponential regression models to identify the model coefficients. A new model is developed in the second phase, integrating the prediction values obtained from the first phase with the GP method. This model is then used to derive the final prediction values. Future load demands are projected up to 2031 based on two different scenarios. The primary contributions of this study are outlined as follows: 1) A novel ensemble method based on GP is introduced for long-term load forecasting in Ethiopia, utilizing GA, PSO, SA, and GP algorithms. These optimization algorithms are used to optimize the parameter coefficients of linear, quadratic, and exponential regression models.

3) The accuracy of the proposed models was evaluated using four statistical metrics: mean absolute percentage error (MAPE), mean square error (MSE), root mean square error (RMSE), and correlation coefficient (R2).

3) Electricity consumption forecasts for 2022–2031 were produced using models under two different scenarios. The first scenario averaged the growth rate of input data from the last 3 years, while the second scenario averaged it over the previous 5 years.

4) The findings suggest that the proposed ensemble method enhances load forecasting accuracy and efficiency. Regular electricity demand assessments remain essential for Ethiopia.

Materials and Methods

This section provides a comprehensive overview of the dataset, optimization methods, regression models, and error metrics employed in the study, presented in sequence.

Dataset

The dataset consists of sixteen instances and includes four independent variables: population, GDP, imports, and exports, along with one dependent variable—energy demand. The values are utilized without any preprocessing, such as normalization, meaning they are used in their raw format. Input data, including GDP, population, and trade data, were selected based on existing literature and domain knowledge, considering their significant relationship with electricity consumption trends. GDP was included as a measure of economic activity, which is strongly correlated with energy demand. Population data was chosen because it directly influences electricity use at both residential and national levels. Trade data, representing industrial and commercial activity, were included to capture the effects of external economic interactions on energy consumption. These data were obtained from international databases (e.g., International Energy Agency, IEA; World Bank, IMF) and reliable sources. Table 1 presents the economic indicators and electricity consumption data of Ethiopia from 2006 to 2021 (International Energy Agency, 2022; The World Bank, 2023).

Table 1 Ethiopia’s electricity consumption, population, GDP, import and export data (International Energy Agency, 2022; The World Bank, 2023).

Year	Electricity consumption (TWh)	Population
(106)	Gross national product
($ 109)	Imports
($ 109)	Exports
($ 109)	
2006	2.94	79.69	15.28	4.13	2.15	
2007	3.19	81.99	19.71	6.07	2.84	
2008	3.29	84.36	27.07	9.01	3.42	
2009	3.29	86.75	32.44	8.35	3.36	
2010	3.86	89.24	29.93	9.12	4.23	
2011	4.46	91.82	31.95	10.08	5.33	
2012	5.29	94.45	43.31	13.7	5.96	
2013	6.41	97.08	47.65	13.81	5.95	
2014	6.62	99.74	55.61	16.18	6.47	
2015	7.67	102.47	64.59	19.57	6.05	
2016	8.66	105.29	74.3	20.12	5.8	
2017	8.86	108.19	81.77	19.2	6.24	
2018	10.36	111.13	84.27	19.24	7.06	
2019	10.68	114.12	95.91	20.02	7.62	
2020	11.11	117.19	107.66	18.17	7.67	
2021	11.58	120.28	111.27	18.54	8.45	

Optimization methods

Optimization algorithms have been successfully used in many real-world problems, such as Energy, renewable energy sources, power systems, and engineering applications, each with different complexity and search space size (Yang et al., 2024a). Hybrid methods combining meta-heuristic algorithms have been developed to improve prediction accuracy and efficiency (Cui et al., 2021). The fundamental details of the optimization algorithms utilized in this study are presented below.

Genetic algorithm

GA is an optimization method based on the evolutionary principle of natural selection. This algorithm simulates the process of natural evolution by enabling convergence to global extrema within a complex solution space. GA offers several advantages: it can be applied to large-scale complex problems, requires fewer control parameters, and operates faster than other algorithms. Three parameters of GA (mutation rate, crossover rate, and number of populations) should be optimized to increase the efficiency of GA or the accuracy of the results obtained (Ladan & Kalantar, 2011).

Particle swarm optimization

PSO is a global optimization algorithm proposed by Kennedy and Eberhart, inspired by the movement of flocks of birds. In this approach, solution candidates are referred to as “particles,” which navigate the solution space in search of the best answer. Within the PSO algorithm, potential solutions generate new possibilities by interacting with their own best solution (the particle’s local best) and the best solution identified by the swarm during iterations. The PSO algorithm operates iteratively, comparing the generated solution at each iteration against both its local best and the swarm’s global best. PSO has proven to be an effective method for addressing long-term load forecasting challenges alongside other approaches (Şahman, Mutluer & Çunkaş, 2022). PSO has proven to be an effective method for addressing long-term load forecasting challenges alongside other approaches.

Simulated annealing

The simulated annealing algorithm (SA) was first introduced by Kirkpatrick, Gelatt & Vecchi (1983). The annealing process inspired the algorithm in metallurgy, where a material is heated and then gradually cooled to remove defects and reduce its energy to reach a minimum energy state (Akram & Habib, 2023; Du, Jin & Zhang, 2016). The algorithm begins with an initial solution and iteratively explores the solution space through minor modifications. It accepts new solutions that improve the objective function value with decreasing probability and occasionally accepts solutions that worsen the objective function value. The “temperature” parameter regulates the likelihood of accepting suboptimal solutions during the early stages of the search (Wang, Zhou & Chen, 2007).

Genetic programming

Michael Cramer initially laid the foundations of GP, which John Koza later expanded. GP is a specialized form of GA, functioning as an evolutionary algorithm designed to optimize computer programs for specific problems. It is often used in the areas of symbolic regression and automatic design. The methodology of GP is based on the principles of natural selection and genetics, similar to those of GAs (Ha et al., 2017). In GP, a population of candidate programs (represented as trees) is developed over generations. The initial population is randomly generated. Then, a process of selection, crossover, and mutation is used to generate new programs, which are then evaluated for their fitness (i.e., their ability to solve the problem). The population evolves to include programs that are better suited to solving the problem. The crossover and mutation operators, which are critical in GP just as they are in GAs, cause changes in the tree structure within the chromosome. These operators not only alter the values within the tree but also modify its shape. GP is an algorithm that seeks to determine both the shape of the model and its coefficients (Karabulut, Alkan & Yilmaz, 2008).

Regression analysis

A variety of regression models exist, and the specific models utilized in this study are presented in Eqs. (1)–(3). Linear, quadratic, and exponential models are well-established in literature as effective methods for estimating energy consumption. The selection of regression models relied on their proven validity in the literature, the specific characteristics of our dataset, and a performance analysis based on optimization results (Özdemir, Dörterler & Aydın, 2022).

Linear regression: The model attempts to predict the dependent variable as a first-order (linear) function of the independent variables. Multivariate linear regression is used because more than one variable affects load prediction.

(1) Y=w0+w1X1+w2X2+⋯+wn∗Xn.

Quadratic regression: The method attempts to predict the dependent variable as a quadratic (square) function of the independent variables.

(2) Y=w0+w1X1+w2X2+⋯wnX1X2+⋯+wmX12+wkX22.

Exponential regression: This method models the data with an exponential function to best fit a trend line.

(3) Y=w0+w1ew2X1+w3ew4X2+⋯+wnewn+1Xm.

The variables in the models are defined as: where Y is the dependent variable (energy consumption), X1, X2, …, Xn are the independent variables (factors population, GDP, import, export etc.) and w1, w2, w3, …, wn are the regression coefficients (coefficients showing the effects of the independent variables).

Performance metrics

In this study, the MSE, RMSE, correlation coefficient (R2), and MAPE performance metrics have been selected. The formulations for these performance metrics are provided in Eqs. (4)–(7) (Özdemir, Dörterler & Aydın, 2022).

(4) MSE=1n∑i=1n⁡(Ai−Fi)2

(5) RMSE=1n∑i=1n⁡(Ai−Fi)2

(6) R2=1−∑i=1n⁡(Ai−Fi)2∑i=1n⁡(Ai−A¯)2

(7) MAPE=1n∑i=1n⁡|Ai−FiAi|

Ai is the actual value for the ith sample, Fi is the forecasted value for the ith sample, A¯ is the mean value of the actual values and n is number of samples.

Hyperparameters

We selected the hyperparameters through an extensive trial-and-error process. First, we determined the hyperparameters’ value ranges and systematically tested different combinations. The results obtained were compared in terms of accuracy, convergence time, and stability. For example, a high mutation rate in GA increased diversity while decreasing stability and a high population size or particle number increased computational time. Finally, we fine-tuned the values that gave the best results. These hyperparameters are detailed in Table 2.

Table 2 Hyperparameters setting for SA, GA, PSO, and GP methods.

Algorithm	Hyperparameters	Linear	Quadratic	Exponential	
GA	Population size	250	250	250	
Crossover fraction	0.8	0.8	0.8	
Crossover function	Dispersed	Dispersed	Dispersed	
Fitness scaling function	Rank scaling	Rank scaling	Rank scaling	
Mutation function	Adaptive feasible	Adaptive feasible	Adaptive feasible	
Selection function	Turnover	Reminder	Turnover	
Maximum number of generations	5,000	25,000	5,000	
Function tolerance	10−6	10−10	10−6	
PSO	Number of particles	200	200	200	
Social Cohesion weight	0.9	0.9	0.9	
Cognitive weight	0.65	0.65	0.65	
Maximum number of generations	10,000	25,000	10,000	
Minimum neigbor ratio	0.4	0.4	0.4	
Function tolerance	10−6	10−10	10−8	
SA	Initial temperature	200	200	200	
Maximum number of iterations	10,000	20,000	10,000	
Function tolerance	10−6	10−6	10−6	
GP	Representation type	Tree-based GP	
Population size	5,000	
Mutation rate	0.1	
Crossover rate	0.9	
Selection function	Tournament selection with a size of 3	
Function set	{+, ×, ÷, −}	
Terminal set	{ Y1 (GA_quadratic),
Y2 (PSO_quadratic),
Y3 (SA_quadratic),
random values in the range of (−2, 2) }	
Maximum number of generations	100	
Function tolerance	0	

Statistical tests

We applied the Wilcoxon signed-rank test to evaluate if there are statistically significant performance differences between model pairs based on their MAPE, using 10 independent sample runs. The p-values between GA-PSO, GA-SA, and PSO-SA are 0.472, 0.156 and 0.160 respectively. This analysis shows that the performance between GA-SA and PSO-SA is slightly different, but the performance of the models is not clearly statistically different. The small statistical difference between the models indicates that the models are relatively close in performance. However, this does not mean that each model does not have advantages for different purposes. The models may be more stable or give better overall results when used together. Therefore, better prediction accuracy was obtained when used with GP.

Genetic programming-based ensemble method

In this study, we propose an ensemble method for forecasting Ethiopia’s annual total energy consumption. Figure 1 illustrates the proposed GP-based ensemble method, which consists of two stages. In the initial stage, the first prediction values are obtained using various optimization techniques and regression models. In the second stage, these initial predictions are refined with the GP method to produce the final forecast value. The model utilizes a range of inputs, including data related to population, GDP, imports, and exports.

Figure 1 Flowchart of the proposed genetic programming-based ensemble method.

Stage-1

In the first stage of the prediction model, coefficients for various regression models were identified using the GA, PSO, and SA methods. In this stage, estimates were generated through nine different approaches: GA_Linear, GA_Quadratic, GA_Exponential, PSO_Linear, PSO_Quadratic, PSO_Exponential, SA_Linear, and SA_Quadratic SA_Exponential.

GA model

The GA method was executed using the parameter settings in Table 2, and the optimal coefficients for linear, quadratic, and exponential models were determined.

The model coefficients and model equations obtained through the GA method are presented in Eqs. (8)–(10).

(8) Linear:Y=0.2878X1−0.0033X2+0.0327X3−0.3784X4−19.8964

(9) Quadratic:Y=0.1626X1+0.0605X2−0.6461X3−0.1311X4−0.0022X1X2+0.0026X1X3+0.0173X1X4−0.0048X2X3−0.0046X2X4+0.2852X3X4+0.00039594X12+0.0018X22−0.022X32−0.4951X42−11.5408

(10) Exponential:Y=−69.197e−0.011X1+409.4632e−0.43X2+592.5662e−60.8343X3+7.5462e−0.8444X4+30.024.

where Y is the yearly energy consumption, X1 is the population, X2 is the GDP, X3 is the import and X4 is the export. Utilizing Eqs. (8)–(10), the electricity consumption for the period 2006–2021 has been computed, with the results presented in Table 3. The RMSE values for the GA_Linear, GA_Quadratic, and GA_Exponential methods are 0.364, 0.312, and 0.349, respectively. Among the GA and regression models, the GA_Quadratic yielded the lowest RMSE value.

Table 3 GA, PSO, and SA forecasts for linear, quadratic, and exponential models (unit:TWh).

Year	Elec. Cons.	GA forecast	PSO forecast	SA forecast	
Line.	Quad.	Exp.	Line.	Quad.	Exp.	Line.	Quad.	Exp.	
2006	2.94	2.31	2.88	3.05	2.36	2.93	2.89	2.28	2.73	2.45	
2007	3.19	2.76	3.1	2.74	2.79	2.84	2.85	2.84	3.08	2.83	
2008	3.29	3.29	3.24	3.11	3.31	3.12	3.02	3.68	3.58	3.33	
2009	3.29	3.96	3.38	3.84	4	3.43	3.19	4.08	3.61	3.24	
2010	3.86	4.39	4.29	4.33	4.37	3.72	3.87	4.07	4.08	3.84	
2011	4.46	4.74	4.4	4.93	4.68	4.24	4.44	4.45	4.62	4.83	
2012	5.29	5.34	5.35	5.61	5.27	5.36	5.23	5.64	5.6	6.06	
2013	6.41	6.09	5.79	6.31	6.03	5.73	5.76	6.01	5.87	6.07	
2014	6.62	6.71	6.8	6.98	6.65	6.79	6.62	6.85	6.84	7.36	
2015	7.67	7.73	7.99	7.67	7.72	7.67	7.71	7.71	8.03	8.47	
2016	8.66	8.62	8.07	8.37	8.65	8.35	8.41	8.49	8.7	8.57	
2017	8.86	9.24	8.96	9.03	9.26	9.05	8.9	9.11	9.07	8.47	
2018	10.36	9.77	9.87	9.68	9.75	9.89	9.57	9.45	9.62	9.62	
2019	10.68	10.4	10.87	10.33	10.39	10.75	10.38	10.5	10.83	11.07	
2020	11.11	11.17	11.08	10.99	11.18	10.82	10.8	11.35	11.03	10.06	
2021	11.58	11.76	11.15	11.62	11.74	11.5	11.59	11.79	11.82	11.85	

PSO model

The algorithm was executed with specific parameters, and optimal coefficients for linear, quadratic, and exponential models were determined. The parameters for the PSO method were set, as shown in Table 2.

The model coefficients and the complete model equation derived through the PSO method are presented in Eqs. (11)–(13).

(11) Linear:Y=0.2886X1−0.0012X2+0.0337X3−0.4287X4−19.8404

(12) Quadratic:Y=−0.0000271146X1−0.0000148687X2−0.97829X3+0.012637X4+0.0000121832X1X2+0.0149X1X3−0.03247X1X4−0.00258X2X3−0.02837X2X4+0.12942X3X4+0.0014X12+0.00122X22−0.0269X32+0.23872X42−2.2864

(13) Exponential:Y= 24.4864e0.0052X1+9.3483e−0.1054X2+3.6346e0.0262X3+2.9082e−2.8138X4−40.0891.

Using Eqs. (11)–(13), the electricity consumption for the period from 2006 to 2021 has been calculated and is presented in Table 3. The RMSE values for the PSO_Linear, PSO_Quadratic, and PSO_Exponential models are 0.363, 0.272, and 0.312, respectively. Among the PSO and regression models, the PSO_Quadratic method yielded the lowest RMSE value.

SA model

The algorithm was run with the specified parameters listed in Table 2, and the most appropriate coefficients were found for the linear, quadratic, and exponential models.

The model coefficients and full model equation obtained by the SA method are given in Eqs. (14)–(16).

(14) Linear:Y=0.0114X1+0.0775X2+0.0541X3+0.1315X4−0.3191

(15) Quadratic:Y=0.0004268X1−0.00055X2+0.00534994X3+0.0011499X4−0.000579X1X2+0.000136X1X3+0.002647X1X4+0.002549X2X3+0.0010495X2X4−0.00105X3X4+0.00041X12+0.000329X22+0.0021499X32−0.005X42+0.00502

(16) Exponential:Y= −0.4123e−0.316X1+0.289e−0.6552X2+0.0921e0.1856X2+0.7385e0.2832X4+0.8932

Electricity energy consumptions between 2006 and 2021 are calculated using Eqs. (14)–(16) and given in Table 3. According to RMSE values, the values of SA_Linear, SA_Quadratic, and SA_Exponential are 0.416, 0.312, and 0.527, respectively. According to the estimates of SA and regression models, the lowest RMSE value was found with the SA_Quadratic method.

The error values associated with all predictions generated at the conclusion of the first stage are presented in Table 4 and Fig. 2, based on the selected evaluation metrics. The best prediction errors for each error criterion are highlighted in bold in the table. Upon examining the error values in the table and the figure, it is observed that the most successful prediction values among the nine results obtained with the proposed method were achieved using quadratic optimization approaches. Although the PSO_Exponential model appears better than the PSO_Quadratic model in terms of the MAPE error criterion, it exhibits worse prediction errors for the MSE and RMSE criteria. When optimization methods are compared, the PSO_Quadratic model outperformed the GA_Quadratic and SA_Quadratic models, achieving a superior RMSE value of 0.272.

Table 4 Forecast errors for linear, quadratic, and exponential models at first stage.

Metrics	GA	PSO	SA	
Lin.	Quad.	Exp.	Lin.	Quad.	Exp.	Lin.	Quad.	Exp.	
Errors obtained from real data	
MSE	0.132	0.097	0.122	0.132	0.074	0.094	0.173	0.098	0.278	
RMSE	0.364	0.312	0.349	0.363	0.272	0.307	0.416	0.312	0.527	
R2	0.986	0.990	0.988	0.986	0.994	0.994	0.981	0.990	0.970	
MAPE	6.16	3.61	5.77	5.95	3.63	3.30	6.94	4.68	6.83	
Errors obtained from test data	
MSE	0.734	0.695	0.674	0.749	0.665	0.796	1.807	1.214	2.015	
RMSE	0.857	0.834	0.821	0.865	0.815	0.892	1.344	1.101	1.419	
R2	0.871	0.908	0.881	0.868	0.897	0.875	0.691	0.789	0.701	
MAPE	7.85	7.64	8.36	7.89	7.86	8.42	11.77	7.86	11.72	
Note:

The best prediction errors for each error criterion are indicated in bold.

Figure 2 MSE, RMSE, R2 and MAPE values for linear, quadratic, and exponential models at the first stage.

Table 4 also presents the error values obtained from using the synthetic data as test data. Synthetic data are also generated to test the model developed using data from 2006 to 2021. The process of synthetic data generation is widely used to enhance datasets and evaluate model performance. In this study, a Gaussian Copula-based method was applied to generate synthetic data. This approach analyzes the probabilistic structure of the original dataset. It accounts for the dependencies among variables to produce new data points with statistical properties similar to those of actual data. As a result, the generated synthetic data serve as an alternative test dataset to assess and validate the model’s performance. Upon examining the results in Table 4, it is observed that quadratic models yield more successful predictions for both real and test data.

As a result, at the end of the first stage, the three best prediction values (GA_Quadratic, PSO_Quadratic, and SA_Quadratic) among these nine results were identified as input values for the second stage. As shown in Table 4, quadratic models consistently yielded lower RMSE, R2, and MAPE values compared to linear and exponential models, indicating their ability to capture underlying patterns in the dataset better. Electricity consumption is influenced by various socioeconomic factors, such as GDP growth, population increase, and industrial expansion, which often exhibit nonlinear relationships with energy demand. The flexibility of the quadratic model enables it to accommodate these nonlinearities more effectively than a purely linear approach.

Conventional curve fitting models

In this study, the conventional curve fitting method, a fundamental approach, was selected as the comparison method to evaluate the predictive performance of the proposed GP-based ensemble method. The curve fitting methods were applied to the same input data. Predictions were also generated using classical linear, conventional quadratic, and conventional exponential regression techniques within the traditional curve fitting framework. The results are presented in Table 5. Based on these prediction outcomes, the RMSE error values for classical linear, classical quadratic, and classical exponential regression were found to be 0.362, 0.471, and 0.376, respectively. It is observed that the second-order models in the first stage have lower prediction errors compared to the curve-fitting method. These findings suggest that the chosen optimization methods yield more successful results compared to classical curve fitting models for these data.

Table 5 Forecast errors for quadratic optimizations and curve-fitting conventional methods.

Metrics	Stage 1 (best results)	Compare methods	
GA_ Quadratic	PSO_ Quadratic	SA_ Quadratic	Curve fitting	
Linear	Quadratic	Exponential	
MSE	0.097	0.074	0.098	0.131	0.222	0.141	
RMSE	0.312	0.272	0.312	0.362	0.471	0.376	
R2	0.990	0.994	0.990	0.986	0.989	0.985	
MAPE	3.61	3.63	4.68	5.97	6.74	5.57	

Stage-2

In the second stage, the forecasted values of GA_ Quadratic, PSO_ Quadratic, and SA_ Quadratic were incorporated into the GP model. By utilizing the optimal forecast values derived from Stage 1, an equation was formulated through GP, leading to the final forecast output.

To determine the parameters of the GP model, including the terminal set, function set, and other elements related to the evolutionary process, we conducted preliminary experiments. At the start of these runs, we added more functions to the function set, including minimum, maximum, square-root, and inverse functions, and employed a relatively wide range of constants in the terminal set. The GP with these settings rarely performed well, but it tended to evolve complex functions. In other words, the GP produced well-performing simple functions when simple functions and a relatively narrow range of constant values were used in its function and terminal sets, respectively. At the end of the preliminary runs, the parameters for the GP optimization method were set as in Table 2. This model is obtained with GP.

(17) F=Y1+0.1−0.1−Y1+(Y2−Y1+2)2+1

where F is the forecast value, Y1 the forecast values of PSO_Quadratic, Y2 the forecast values of SA_Quadratic. According to the formula obtained with GP, the forecast values of GA_Quadratic are not used. In this case, the model might have chosen only those inputs that demonstrate the highest predictive accuracy or contribute most significantly to the target variable.

The error values corresponding to the optimal results from Stage 1, as well as the error values of the final predictions derived from Stage 2 of the proposed method, are presented in Table 6 and Fig. 3. According to all error metrics, the predictions resulting from Stage 2 are the most accurate. The estimation errors obtained in Stage 2 were found to be 0.039, 0.198, 0.995, and 2.83 for MSE, RMSE, R2, and MAPE respectively. Furthermore, the final prediction results obtained by the proposed ensemble method outperform those obtained in Stage 1 across all metrics.

Table 6 Error comparison for quadratic models and proposed method.

Metrics	Stage 1 (best results)	Stage 2	
GA_ Quadratic	PSO_ Quadratic	SA_ Quadratic	Proposed ensemble method	
MSE	0.097	0.074	0.098	0.039	
RMSE	0.312	0.272	0.312	0.198	
R2	0.990	0.994	0.990	0.995	
MAPE	3.61	3.63	4.68	2.83	
Note:

The best prediction errors for each error criterion are indicated in bold.

Figure 3 Performance metrics for quadratic models and proposed method.

When the model’s performance is examined on an annual basis, the error values are relatively low in some years. The model can make predictions with higher accuracy, especially in periods where the data set does not show rapid change. On the other hand, there has been a slight increase in errors during periods of rapid economic growth or sudden changes. This situation indicates that predictions can be more difficult in years of radical change.

Parameters sensitivity analysis and confidence interval

Parameters sensitivity analysis

In Table 7 and Fig. 4, the correlation relationships between the input parameters (X1, X2, X3, X4) and the output variable (Y) are analyzed based on the results of the Monte Carlo analysis. The standard deviation and variance values express the width and variability of the distribution of each parameter. As can be seen in the table and figure, parameter X1 is a factor that significantly affects the output (Y) of the system, as it exhibits both high correlation coefficients and a balanced variance. Therefore, X1 should be regarded as a priority variable in model improvement and optimization processes. The low correlation coefficients and variability of X2 and X4 indicate that the effect of these parameters on Y is weak. X3 holds some importance in terms of quadratic relationships, but it is low in other types of relationships and variability. This parameter can be considered as a second priority. The input variables X1, X2, X3, and X4 denote population, GDP, imports, and exports, respectively.

Table 7 The correlations and variances between the input parameters.

Input parameters	Correlation	Standart deviation	Variation	
Linear	Quadratic	Exponential	
X1	0.97524	0.6439	0.9936	11.958	143.012	
X2	0.0049	−0.0627	0.0253	27.069	732.775	
X3	0.0259	0.2711	−0.0054	4.57	20.886	
X4	−0.1716	−0.1116	−0.0366	1.78	3.171	

Figure 4 The correlations between the input parameters.

Confidence interval

Figure 5 shows the confidence interval analysis of the GP-based ensemble method. The estimated consumption data and the actual consumption data are within the 95% confidence interval limits. Considering the strong correlation between the estimated consumption values and the actual consumption values, it is possible to accurately estimate the consumption values with the intended GP-based ensemble method.

Figure 5 The confidence interval analysis of the GP-based ensemble method.

Results

Two scenarios were developed to assess the impact of increases in future input parameter variables on electricity energy consumption. In Scenario 1, the growth rates of the last 3 years are averaged, reflecting a short-term trend that captures more recent dynamics and possible fluctuations. Scenario 2, on the other hand, provides a broader and more fluid historical perspective that considers medium-term patterns, using average growth rates over the last 5 years.

Scenario 1

In Scenario 1, the growth rate of the input parameters over the last 3 years is averaged. Population, GDP, imports, and exports are assumed to increase by 2.67%, 9.81%, 2.03%, and 6.25%, respectively. Between the years 2022 and 2031, the input parameters are increased each year according to Scenario 1, and using this data, the annual electrical energy consumption is estimated based on the proposed ensemble method. According to Scenario 1, the values of the input parameters for the period from 2022 to 2031 are presented in Table 8. Additionally, Table 8 includes the forecasts generated by the GA_Quadratic, PSO_Quadratic, and SA_Quadratic methods, which were identified as the most effective in Stage 1. Based on these forecasts, Ethiopia’s annual electricity consumption is projected to range from 11.83 to 40.08 TWh between 2022 and 2031.

Table 8 Forecasts of quadratic models for GA, SA, and PSO at Scenario 1 and Scenario 2.

Year	Scenario 1 [TWh]	Scenario 2 [Twh]	
GA_Quad	PSO_Quad	SA_Quad	GA_Quad	PSO_Quad	SA_Quad	
2022	11.83	12.32	13.05	11.72	12.52	13.15	
2023	12.79	13.27	14.43	12.39	13.71	14.64	
2024	13.98	14.33	16.03	13.01	15.03	16.36	
2025	15.48	15.55	17.9	13.55	16.51	18.32	
2026	17.4	16.98	20.08	14.04	18.17	20.58	
2027	19.86	18.45	22.65	14.38	20.06	23.18	
2028	23.13	20.69	25.65	14.62	22.2	26.18	
2029	27.33	23.17	29.2	14.65	24.64	29.65	
2030	32.86	26.24	33.41	14.47	27.44	33.67	
2031	40.08	30.07	38.4	14.02	30.68	38.33	

Scenario-2

In Scenario 2, the growth rates of the parameters over the last 5 years are averaged. Population, GDP, imports, and exports are projected to grow by 2.7%, 8.51%, 4.05%, and 9.9%, respectively. Between 2022 and 2031, the input parameters were increased each year according to Scenario 2, and using this data, the annual electricity consumption was estimated with the proposed ensemble method. According to Scenario 2, the input parameter values for the period from 2022 to 2031 are presented in Table 8. The methods identified as the most effective in Stage 1—GA_Quadratic, PSO_Quadratic, and SA_Quadratic—are also detailed in Table 8. Based on this data, Ethiopia’s annual electricity consumption is projected to range from 11.72 to 38.33 TWh during the period from 2022 to 2031.

Discussion

This section discusses the implications of the results obtained in the ‘Results’ section and future forecasts of the ensemble method. Figure 6 illustrates the future predicted values using quadratic methods in Stage 1 for both scenarios. Based on the forecast values derived from these methods, the SA_Quadratic method yields comparable results in both scenarios. For the year 2031, electricity consumption is projected to be approximately 38 TWh. According to the PSO_Quadratic method, electricity consumption for 2031 is estimated to be around 30 TWh, although somewhat similar predictions are obtained in both scenarios. The GA_Quadratic model provides significantly different results in each scenario compared to the other methods. In Scenario 1, the GA_Quadratic result was about 40 TWh for 2031, while in Scenario 2, the electricity consumption was around 14 TWh for the same year. When all these results are analyzed collectively, the quadratic models that produced the best results in Stage 1 yielded quite different prediction values when considered independently. This substantial discrepancy is unacceptable for an issue like annual energy consumption, which influences the rate at which the country’s energy infrastructure expands. Therefore, rather than relying on individual methods in the first stage, it is more appropriate to utilize the proposed GP-based ensemble method, which models the data from 2006 to 2021 with the lowest error, for forecasting future outcomes years.

Figure 6 Future electricity consumptions for Scenario 1 and Scenario 2 using quadratic models.

The input values in Scenario 1 and Scenario 2 were updated and applied to the ensemble method proposed in this study. The forecasts for the ensemble method are shown in Table 9. According to this forecast, Ethiopia’s annual electricity consumption between 2022 and 2031 ranges from 12.45 to 30.17 TWh for Scenario 1 and 12.64–30.78 TWh for Scenario 2. According to the rates of change of the input variables determined in Scenario 1 and Scenario 2, electricity consumption is expected to be higher in Scenario 2 than in Scenario 1. While electricity consumption in 2021 was 11.58 TWh, it can be seen that the consumption value in both scenarios will exceed 30 TWh. According to these values, about three times more electricity will be consumed in 2031 compared to 2021. This increase in electricity consumption will require the expansion of generation resources, transmission and distribution lines, and the commissioning of new infrastructure.

Table 9 Future electricity consumptions for Scenario 1 and Scenario 2 using the proposed method [TWh].

Year	Scenario 1	Scenario 2	
2022	12.45	12.64	
2023	13.41	13.83	
2024	14.15	15.16	
2025	15.63	16.71	
2026	17.07	18.23	
2027	18.55	20.15	
2028	20.79	22.29	
2029	23.27	24.74	
2030	26.34	27.54	
2031	30.17	30.78	

The findings of this study have significant implications for Ethiopia’s energy infrastructure planning. The expected tripling of electricity consumption by 2031 highlights the need for strategic investments in power generation, transmission, and distribution systems. To meet this increasing demand, Ethiopia must prioritize expanding the capacity of renewable sources such as hydro, solar, and wind power while also focusing on reducing losses and enhancing reliability through the modernization of the existing grid. Encouraging public-private partnerships is advisable to establish a robust energy infrastructure that can satisfy Ethiopia’s future energy requirements and bolster economic growth. Furthermore, incorporating the proposed community forecasting model into policy planning can enhance decision-making processes and facilitate optimal decision-making planning. This article also has some limitations. The number of samples in the data set is very small (16), which partly weakens the generalization ability of the model. Another limitation is that it is impossible to validate the forecasting performance of the model in advance for future years when the input parameters of the model change unpredictably depending on many factors. These limitations should be considered when interpreting the results. Despite these limitations, the findings show that ensemble methods can improve forecast accuracy and provide a projection for energy planning.

Conclusions

This study presents an ensemble method based on GP that integrates SA, GA, and PSO algorithms to enhance long-term load forecasting. The model was tested on historical data and synthetic data, demonstrating its potential to produce reasonable electricity consumption estimates. After the model’s accuracy was validated, Ethiopia’s total annual electricity consumption was estimated up to 2031 based on two scenarios. According to the forecast for electric power consumption in the coming years, electricity usage in 2031 is expected to increase by about threefold compared to 2021 in both case scenarios.

This article highlights the advantages of the proposed ensemble method, which combines SA, GA, PSO, and GP algorithms, supported by a series of statistical methods tests. Although the model shows accuracy based on historical data and synthetic data, the small dataset size limits the model’s generalization ability. Future research should focus on expanding the dataset, including additional influential variables, and using appropriate validation techniques to improve model reliability. Despite these limitations, the findings suggest that ensemble methods show promise in improving long-term electricity demand forecasting and provide a helpful tool for policymakers and energy planners.

Supplemental Information

Supplemental Information 1 Ethiopia’s electricity consumption, population, GDP, import and export data.

Supplemental Information 2 Python Code for Ensemble Method.

Supplemental Information 3 Linear model.

Supplemental Information 4 Exponential model.

Supplemental Information 5 Quadratic model.

During the preparation of this work, the author(s) utilized ChatGPT to assist with language refinement, grammar correction, and enhancing the clarity of the text. After using this tool, the author(s) thoroughly reviewed and edited the content to ensure accuracy, technical correctness, and alignment with the research objectives. The author(s) take full responsibility for the final content of this publication.

Additional Information and Declarations

Competing Interests

Mehmet Cunkas is an Section Editor for PeerJ computer science

Author Contributions

Hayat Ahmed Issa conceived and designed the experiments, performed the experiments, performed the computation work, prepared figures and/or tables, and approved the final draft.

Hasan Hüseyin Çevik conceived and designed the experiments, performed the experiments, analyzed the data, performed the computation work, prepared figures and/or tables, authored or reviewed drafts of the article, and approved the final draft.

Ahmet Yilmaz conceived and designed the experiments, performed the experiments, analyzed the data, performed the computation work, authored or reviewed drafts of the article, and approved the final draft.

Mehmet Cunkas conceived and designed the experiments, analyzed the data, authored or reviewed drafts of the article, and approved the final draft.

Data Availability

The following information was supplied regarding data availability:

The code and data are available in the Supplementary Files.

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
