# Peer review of "A genetic programming-based ensemble method for long-term electricity demand forecasting"

_PeerJ Computer Science, doi:10.7717/peerj-cs.2825_

## Round 0.1 · original submission · Major Revisions

· Academic Editor

Major Revisions

Dear authors,

Reviewers have now commented on your paper. You will see that they advise you to make major revisions to your manuscript. If you are prepared to undertake the work required, I would be pleased to reconsider my decision.

If you decide to revise the work, please submit a list of changes or a rebuttal against each point that is being raised when you submit the revised manuscript.

Best wishes,
D. Pamucar


Reviewer 1 ·

Basic reporting

see attached file.

Experimental design

see attached file.

Validity of the findings

see attached file.

Additional comments

see attached file.

Annotated reviews are not available for download in order to protect the identity of reviewers who chose to remain anonymous.

·

Basic reporting

Strengths:

-The manuscript is written in clear, professional English
-Introduction effectively establishes context with relevant background on Ethiopia's energy situation
-Literature is well-referenced, with comprehensive coverage of both local Ethiopian studies and broader forecasting methodologies
-Figures and tables are relevant, properly labeled, and support the findings
-Structure follows standard research paper format

Areas for improvement:

-Some equations contain formatting issues that should be cleaned up for clarity (e.g., equations 8-10)
-A few typos/formatting inconsistencies need to be addressed throughout
-Consider adding a list of abbreviations for the various algorithms (GA, PSO, SA, etc.)

Experimental design

Strengths:

-Research objectives are clearly defined with identified knowledge gaps
-Methodology is rigorous with detailed explanation of the two-stage ensemble approach
-Multiple algorithms and regression models are systematically compared
-Input parameters and modeling assumptions are well justified
-Two different scenarios provide valuable comparative analysis

Areas for improvement:

-More details needed on parameter selection for the algorithms (e.g., why specific values were chosen)
-Additional justification for the choice of input variables would strengthen the methodology
-Consider discussing potential limitations of the proposed approach
-Include more details on computational requirements and processing time

Validity of the findings

Strengths:

-Results are thoroughly validated using multiple error metrics (MAPE, MSE, RMSE, R²)
-Comparative analysis between different methods is comprehensive
-Findings are well-supported by the data and analysis
-Conclusions directly address the research objectives
-Future projections are presented with appropriate context and implications

Areas for improvement:

-Include confidence intervals or uncertainty measures for the forecasts
-Provide more discussion on why certain methods performed better than others
-Consider sensitivity analysis of key parameters
-Add more discussion on practical implications for energy planning

Additional comments

-The paper makes a valuable contribution to long-term electricity demand forecasting methodology
-The focus on Ethiopia provides important insights for developing nations
-The ensemble approach demonstrates clear advantages over individual methods
-Consider expanding discussion on potential policy implications
-Future work could explore additional scenarios and variables

Reviewer 3 ·

Basic reporting

1. The author may strengthen the introduction by providing an extended explanation of the difficulties involved in long-term electricity demand forecasting in developing countries, especially those with growing economic rates of growth and expanding urbanization such as Ethiopia. More development on the specific challenges that Ethiopian is facing would have given stronger reason for the study. Could you give us more instances on how accurate forecasting will assist in planning, formulation of policies and arrangements for energy facilities in Ethiopia?

2. It may also be useful to draw a more detailed comparison of the proposed method with other ensemble learning models employed in the field of electricity demand forecasting. Moreover, a better understanding of the areas within a more comprehensive literature review where previous research has shortcomings concentrating on the energy demand in Ethiopia and how this paper tries to address those shortages would enhance the paper.

3. Some inconsistency in naming conventions is observed, for instance, different naming for equivalent mathematical operations “GA_Quadratic” and “GA_Squared”. Kindly make sure that the same terms are used throughout the manuscript. Some of them contain unclear constructions which could be formulated better: for instance, the note on page 14, line 364, about the GA_Quadratic method.

4. It will be appreciated if each factual statement made in this report is substantiated by the necessary references. For instance, the points mentioned at page 5, line 35, about electrical consumption in the developed countries and at page 6, line 63, on the expansion of the economy of Ethiopia need citation.

5. Give more information about preprocessing if done on the data alongside with missing values and outliers to make the readers understand from where one is starting the analysis.

Experimental design

1. More justification is required for the incorporation of the selected input variables namely population, GDP, import and export. The reader should explain why these particular variables are chosen before the study and argue any limitation of such variables and other possible variable that could be used.

2. The decision on which specific regression models (linear, quadratic, exponential) to use should be more clearly articulated for the first stage of the ensemble method.

3. What is rationale for using the mean forecast values from first stage as inputs in second stage GP model? What can this approach offer as its benefits?

4. The parameters of the GP model used for training are given in the paper, however there is no detail about the criteria of their selection and how their choice may influence the results of the model. Can you specify further how you came up with your choices of the function set, the terminal set and the crossover/mutation rate?

5. To enhance the credibility of the proposed method additional comparisons with other typical ensemble methods like bagging or boosting should be made.

Validity of the findings

1. It is necessary to analyze how sensitive the model is and analyze the relative importance of input variables. Kindly perform sensitivity analysis to enable determine the impact of changes in the input variables (e.g., the rate of increase in population, the GDP changes).

2. The quadratic regression model fared better than the linear and exponential models. Could you please provide more details about the possible causes of such a superior performance? There is a need to know whether the observed result is a wakeup call due to the nature of data collected, the inherent characteristics of the Ethiopian environment, or any other reasons.

3. Although the paper gives the error values, it is necessary to have a deeper discussion about these values. Examine the findings of the error values in each of the stages and identify any pattern or observation drawn from the study. Is the model good on certain year(s) or certain scenario?

4. For additional measures of uncertainty around the forecasts, it is suggested to present the confidence intervals of the predictions.

5. Create a table or figure that compares the forecast results of the proposed ensemble method with those of the models used in the first stage. This will clearly show the benefit of using the ensemble approach in the current hybrid environment.

Additional comments

1. The discussion section should also be lengthened and contain a broader analysis of the implications of the study’s findings to policy makers and energy planners or any other stakeholder involved in the energy sector in Ethiopia. How does it help decision makers and planners to make decision and allocate resources?
2. It is also recommended to mention the possible difficulties when applying the proposed method in a real environment. As you have pointed out, is it all right to use it or are there any questions concerning the data availability or computation problems to limit its applicability?

3. The ideas for further research are well presented, however they can be expanded. For instance while debating on integrating climate variables, it will be appropriate to identify the most suitable climatic parameters that could be included in the model; and how such parameters can best be incorporated into the model.

4. It may be useful to offer a flowchart or a diagram to represent the general approach in order to shed light on the relationships between different steps and methods. This would improve the understanding of the reader on the proposed ensemble method for developing the model.

---

## Round 0.2 · Major Revisions

· Academic Editor

Major Revisions

Dear authors,

Reviewers have now commented on your paper. You will see that they advise you to make major revisions to your manuscript. If you are prepared to undertake the work required, I would be pleased to reconsider my decision.

If you decide to revise the work, please submit a list of changes or a rebuttal against each point that is being raised when you submit the revised manuscript.

Best wishes,
D. Pamucar

Reviewer 1 ·

Basic reporting

The authors have addressed all my concerns.

Experimental design

na

Validity of the findings

na

Additional comments

na

Reviewer 3 ·

Basic reporting

1. The authors did not split the data into training and testing sets due to the small dataset size (16 instances). This is a critical flaw that undermines the validity of the results. The model's generalization ability cannot be assessed, and the reported performance is likely overoptimistic.
2. The rationale for the specific two-stage ensemble structure (GA, PSO, SA in Stage 1; GP in Stage 2) is weak. Why these particular algorithms? Why GP for combining outputs?
3. The description of hyperparameter selection is vague ("preliminary experiments"). The search space and selection criteria are not clearly defined.
4. The manuscript lacks statistical tests to determine if performance differences between models are statistically significant.

Experimental design

1. The Results section includes interpretation and discussion, which should be separated.
2. The subplots in Figure 3 should have consistent y-axis scales to facilitate direct comparison.
3. Data comes from IEA and the World Bank, cited, but not directly supplied.

Validity of the findings

1. While generally good, the English language needs further polishing. Several sentences lack clarity and precision.
2. The conclusions are overstated given the methodological limitations (especially the lack of data splitting).
3. The justification for the superior performance of the quadratic model is weak and relies on general statements.
4. It is recommended to put the research question at the end of introduction.

Additional comments

N/A

---

## Round 0.3 · accepted · Accept

· Academic Editor

Accept

All the reviewers' comments have been addressed carefully and sufficiently, the revisions are rational from my point of view, I think the current version of the paper can be accepted.

Reviewer 1 ·

Basic reporting

It's okay.

Experimental design

It's okay.

Validity of the findings

It's okay.

Additional comments

n/a

·

Basic reporting

The manuscript presents a genetic programming-based ensemble method for forecasting Ethiopia's long-term electricity consumption. Overall, the reporting quality is satisfactory with appropriate improvements made based on previous reviewer feedback.

Strengths:

-The manuscript is written in clear, professional English with appropriate technical language.
-The literature review adequately establishes the importance of electricity demand forecasting for Ethiopia and situates the work within existing research.
-The authors have effectively highlighted the knowledge gap in applying optimization methods to Ethiopia's energy forecasting.
-The article structure follows standard scientific presentation with clear sections for methodology, results, and discussion.
-Figures and tables are well-presented, with improvements made as requested by previous reviewers (e.g., consistent y-axis scales in Figure 3).
-The research questions have been added to the introduction as requested.
-The manuscript is self-contained and presents a complete body of work without unnecessary subdivision.

Areas for improvement:

-While many citations are provided, some additional references on recent ensemble forecasting methods could further strengthen the background.
-The authors could more explicitly state which table contains the raw data used in their analysis for better transparency.

Experimental design

The experimental design is sound and has been improved with the addition of statistical tests and synthetic data validation as requested by previous reviewers.

Strengths:

-The research question is clearly defined with two fundamental questions now added to the introduction.
-The methodology is rigorous, using a two-stage ensemble approach with multiple optimization algorithms.
-The authors have added explanations for hyperparameter selection through extensive trial-and-error processes.
-The use of synthetic data to address the small dataset limitation (16 instances) demonstrates good research practice.
-The addition of Wilcoxon signed-rank test provides statistical validation of model performance differences.
-The methods are described with sufficient detail to allow replication by other researchers.

Areas for improvement:

-While the authors now mention testing with synthetic data, more details on the specific Gaussian Copula-based method used to generate this data would be helpful.
-The rationale for specific algorithm selection (GA, PSO, SA, GP) has been addressed, but could be further strengthened with justification from performance benchmarks in similar domains.

Validity of the findings

The manuscript presents valid findings with appropriate limitations now acknowledged.

Strengths:

-The data analysis is sound and the results are clearly presented.
-The authors now explicitly address the small dataset size limitation (16 instances) and its potential impact on generalization ability.
-Test data validation using synthetic data strengthens the robustness of the findings.
-Error metrics (MSE, RMSE, R², MAPE) are appropriate for evaluating the model performance.
-Statistical tests have been added to validate performance differences between models.
-The conclusions are appropriately connected to the original research questions.
-The authors have revised overstated conclusions to better reflect the methodological approach and limitations.

Areas for improvement:

-While the synthetic data testing is a positive addition, the authors could provide more details on how well the model performs on completely new, unseen data patterns.
-The discussion on quadratic models' superior performance could further explore theoretical explanations beyond empirical results.

Additional comments

This manuscript makes a valuable contribution to the field of electricity demand forecasting for Ethiopia. The authors have been responsive to previous reviewer feedback and have substantially improved their work accordingly.

Particularly commendable aspects include:

-The clear separation of Results and Discussion sections as requested by reviewers.
-The addition of statistical tests to validate model performance differences.
-The use of synthetic data to address the small dataset limitation.
-More tempered conclusions that acknowledge the limitations of the study.
-The detailed explanation of hyperparameter selection.

The authors' GP-based ensemble approach demonstrates promising results for improving forecast accuracy. The model's ability to project Ethiopia's electricity consumption up to 2031 under two different scenarios provides valuable insights for energy planning and policy-making.

Future research might benefit from:

-Expanding the dataset with additional historical data or more input variables.
-Exploring other ensemble techniques beyond GP.
-Developing methods to better handle rapid changes in economic factors affecting electricity consumption.

Overall, the manuscript represents a solid contribution to the field and, with the improvements made in response to reviewer feedback, is suitable for publication.

Reviewer 3 ·

Basic reporting

The authors did all required revisions. I have no new comments.

Experimental design

N/A

Validity of the findings

N/A